# The Maternal–Neonatal Outcomes of Twin Pregnancies with Preeclampsia and Their Association with Assisted Reproductive Technology: A Retrospective Study

**DOI:** 10.3390/diagnostics12061334

**Published:** 2022-05-27

**Authors:** Huanrong Li, Meilu Lyu, Ruqian Zhao, Yuqin Zang, Pengzhu Huang, Jie Li, Ye Yan, Yingmei Wang, Zengyan Li, Cha Han, Fengxia Xue

**Affiliations:** 1Department of Obstetrics and Gynecology, Tianjin Medical University General Hospital, Tianjin 300052, China; lihuanrong@tmu.edu.cn (H.L.); lyumeilu@tmu.edu.cn (M.L.); zhaorq1994@tmu.edu.cn (R.Z.); zangyq1993@tmu.edu.cn (Y.Z.); hpz1993@tmu.edu.cn (P.H.); cheeseteddy@tmu.edu.cn (J.L.); yanye@tmu.edu.cn (Y.Y.); wangyingmei@tmu.edu.cn (Y.W.); 2Tianjin Key Laboratory of Female Reproductive Health and Eugenic, Tianjin Medical University General Hospital, Tianjin 300052, China

**Keywords:** assisted reproductive technology, outcomes, preeclampsia, twin pregnancies

## Abstract

Objective: This study aimed to investigate the maternal–neonatal outcomes of twin pregnancies of mothers with preeclampsia and their association with assisted reproductive technology (ART). Methods: A retrospective study on the clinical and maternal–neonatal outcome data of 698 women with twin pregnancies who delivered in our hospital from December 2013 to September 2021 was conducted. Continuous variables were analyzed using a Student’s *t*-test or Wilcoxon rank-sum test. Categorical variables were analyzed using the Chi-square test. The risk factors of twin pregnancies with preeclampsia were analyzed by logistic regression. Results: The rate of twin pregnancy complicated by preeclampsia was 17.62% (123/698). Logistic regression analysis showed that ART increased the risk of preeclampsia in twin pregnancies (AOR: 1.868, 95% CI: 1.187–2.941). Mothers with preeclampsia carrying twins conceived with ART had a higher rate of delivery at gestational week < 34 (29.9% vs. 12.5%) and asphyxia of the neonate at 5 min after delivery (13.4% vs. 1.8%) than those with preeclampsia conceived without ART (*p* < 0.05). Conclusions: ART increases the risk of preeclampsia in twin pregnancies and the rate of adverse maternal–neonatal outcomes for twin pregnancies with preeclampsia. The policy of single embryo transfer is a method to reduce the adverse pregnancy outcomes of ART.

## 1. Introduction

In recent years, the incidence of twin pregnancies has increased. Although a small proportion of twin pregnancies account for the total number of pregnancies, the maternal and infant incidence rate of twin pregnancies is higher than that of singleton pregnancies. From 1980 to 2014, the rate of twin pregnancy in the United States increased by 79% [1,2], which is mainly related to the increase in assisted reproductive technology (ART) applications [3]. Although ART has enabled many infertile couples to have a baby, numerous studies have demonstrated that the risk associated with ART is clearly higher than without ART and includes not only twin or multiple pregnancies but also adverse perinatal outcomes, such as hypertensive disorders, postpartum hemorrhage, preterm birth, and birth defects [4,5,6,7,8,9]. Finding out the relationship between ART and pregnancy complications of twin pregnancy is of great significance to avoid adverse pregnancy outcomes and improve the quality of offspring.

Over the past 20 years, the incidence rate of preeclampsia has increased by 25%, and this change has contributed to an annual increase in maternal morbidity and mortality rates worldwide [10,11,12]. The increase in the incidence of preeclampsia is related not only to the application of ART, but also to twin pregnancies and multiple pregnancies. Compared with natural conception, pregnancies conceived with ART have an increased prevalence and risk of preeclampsia [13]. Compared with singleton pregnancies, the prevalence and risk of preeclampsia in twin pregnancies are increased [14]. The prevalence and risk of preeclampsia in women with twin pregnancies who conceived with ART were higher than those in women with twin pregnancies who conceived with spontaneous conception [15]. Many studies have shown that ART increases the rate and risk of preeclampsia in women with twin pregnancies. However, few studies have evaluated the influence of ART on the maternal–neonatal outcomes in women with twin pregnancies complicated by preeclampsia. The relationships between maternal–neonatal outcomes of twin pregnancies, preeclampsia and ART are still not clear. Hence, the purpose of this study was to research the maternal and neonatal outcomes of ART-associated twin pregnancies and twin pregnancies complicated by preeclampsia.

## 2. Materials and Methods

We retrospectively analyzed 698 twin pregnancies from December 2013 to September 2021 at the Gynecology and Obstetrics Department of Tianjin Medical University General Hospital. This hospital provides a full range of services for all people, as well as specialist care for women with high-risk pregnancies. This project was approved by the Ethics Committee of Tianjin Medical University General Hospital (Ethical No. IRB2021-WZ-049) in accordance with the Declaration of Helsinki.

Women with deliveries at <20 weeks of gestation, incomplete outcome data, selected fetal reductions, and termination of pregnancy were excluded. In total, 698 twin pregnancies were finally included. Additionally, all mothers with twin pregnancies delivered once. Among them, 123 women had preeclampsia during pregnancy. We grouped women with twin pregnancies and women with twin pregnancies complicated by preeclampsia into two pairs of groups according to whether those women underwent ART: (1) twin pregnancies conceived with ART and twin pregnancies conceived without ART; (2) twin pregnancies with preeclampsia conceived with ART and twin pregnancies with preeclampsia conceived without ART. ART mainly refers to in vitro fertilization and embryo transfer techniques.

Medical records were collected for further statistical analysis, including basic demographic characteristics, marital and reproductive history, medical history, and maternal–neonatal outcomes. Records were double-checked by a trained obstetrician to avoid mistakes. All diagnoses of diseases were performed according to the 10th edition of the International Classification of Diseases.

The gestational age of pregnancy was corrected according to the last menstrual period of mothers with regular menstrual cycles who conceived spontaneously and the crown-rump lengths of the fetuses via Doppler ultrasonography during the 11–13 + 6 weeks of conception by ART or mothers with irregular menstrual cycles who conceived spontaneously [16]. The chorionic properties were identified according to the number of placenta, lambda and T signs of the membrane, and intertwin membrane thickness upon ultrasound in the first trimester was confirmed by histopathological examination after delivery [17].

Maternal outcomes included the following: intensive care unit (ICU) admission; gestational hypertension [10]; preeclampsia; chronic hypertension; chronic hypertension with superimposed preeclampsia; eclampsia; heart failure; delivery at a gestational age < 37 weeks; delivery at a gestational age < 34 weeks; delivery at a gestational age < 32 weeks; delivery at a gestational age < 28 weeks; placenta previa; placenta accrete; placental abruption; postpartum hemorrhage; gestational diabetes mellitus (GDM); premature rupture of membranes (PROM); preterm premature rupture of membranes (PPROM); hyperthyroidism; hypothyroidism; and hemolysis, elevated liver enzymes, and low platelet count (HELLP) syndrome.

Neonatal outcomes included admission to the neonatal intensive care unit (NICU), low birthweight (birthweight < 2500 g), very low birthweight (birthweight < 1500 g), asphyxia (Apgar score < 7 at 1 min and 5 min after birth), twin-to-twin transfusion syndrome, congenital anomalies, and perinatal mortality.

Gestational hypertension refers to hypertension that occurs after 20 weeks of gestation and is complicated by systolic pressure ≥ 140 mmHg and/or diastolic pressure ≥ 90 mmHg, which occurs without proteinuria and disappears within 12 weeks after delivery [10]. Preeclampsia also occurs after 20 weeks of gestation, with systolic pressure ≥ 140 mmHg and/or diastolic pressure ≥ 90 mmHg, with or without severe visceral damage (thrombocytopenia, impaired liver and kidney function; pulmonary edema; new-onset central nervous system abnormalities; and dysopia). Chronic hypertension refers to hypertension that occurs before 20 weeks of gestation. Chronic hypertension with superimposed preeclampsia includes chronic hypertension with complicating preeclampsia. Eclampsia, resulting from preeclampsia, is a severe complication of hypertensive disorders during pregnancy. Heart failure refers to New York Heart Association classes III or IV. GDM refers to an abnormal oral glucose tolerance test during 24–28 gestational weeks. Elderly primipara was defined as a maternal age of ≥35 years and primiparity. Abortion includes spontaneous abortion (miscarriage) as well as therapeutic abortion. Placenta previa refers to the finding of the placenta over or close to the cervix before delivery by ultrasound. Placenta accreta refers to the placental implantation of the myometrium diagnosed by ultrasound before delivery and by histopathology after delivery. Placental abruption refers to the placenta peeling off from the uterine wall after 20 weeks of gestation before delivery. PROM was defined as fetal membrane rupture before labor, and PPROM was defined as fetal membrane rupture before 37 weeks.

Based on the documented prevalence of twin pregnancies with preeclampsia in the literature of 7.1–14.8% [15,18,19], the sample size was calculated using PASS software, version 15 (NCSS, Kaysville, UT, USA). Setting the prevalence of twin pregnancies with preeclampsia at 10%, considering a two-sided confidence level of 95% and a precision of 0.05 at procedures of confidence intervals for one proportion, we calculated that the minimum sample size of twin pregnancies we needed was 593.

We performed the analysis using SPSS Statistics software, version 22 (IBM, Armonk, NY, USA). Continuous variables were analyzed with a Student’s *t*-test. We used the Chi-square test, continuity correction, and Fisher’s exact test to analyze categorical variables. Logistic regression analysis was used to identify risk factors associated with twin pregnancies. For the adjusted odds ratio (AOR) of twin pregnancies with preeclampsia, we adjusted maternal years, age, gravidity, parity, abortion, polycystic ovary syndrome (PCOS), chorionicity, GDM, hyperthyroidism, and hypothyroidism. A result with *p* < 0.05 was considered significant.

## 3. Results

From December 2013 to September 2021, 29,610 pregnant women delivered in our hospital; among them, there were 698 twin pregnancies, with a twin pregnancy prevalence rate of 2.36% (698/29,610). There were 123 twin pregnancies with preeclampsia, and the prevalence rate of twin pregnancies with preeclampsia was 17.62% (123/698) (Table 1).

Due to the adverse effects of preeclampsia on women with twin pregnancies, we analyzed risk factors that lead to the increased incidence of preeclampsia. Logistic regression analysis found that ART was a risk factor (AOR: 1.868, 95% CI: 1.187–2.941) leading to preeclampsia in twin pregnancies when we adjusted for years, maternal age, gravidity, parity, abortion, PCOS, chorionicity, GDM, hyperthyroidism, and hypothyroidism (Table 2 and Figure 1).

To better illustrate the influence of ART on preeclampsia, we first grouped twin pregnancies into the ART group and the without ART group (Table 3). Among 698 twin pregnancies, 45.99% of women conceived with ART (321/698), and 54.01% of women conceived without ART (377/698). We found that twin pregnancies conceived with ART had a higher rate of maternal age ≥ 35 (38.3% vs. 16.4%, *p* < 0.05), longer inpatient days (6.80 ± 3.925 vs. 6.09 ± 3.273, *p* < 0.05), a higher rate of primiparity (89.7% vs. 77.5%, *p* < 0.05), a higher rate of elderly primipara (32.7% vs. 8.5%, *p* < 0.05), a higher rate of GDM (20.9% vs. 14.3%, *p* < 0.05), and a higher rate of preeclampsia (20.9% vs. 14.9%, *p* < 0.05) (Figure 2), while a lower rate of monochorionic twin pregnancies (4.4% vs. 19.1%, *p* < 0.05) than twin pregnancies were conceived without ART. Maternal and neonatal outcomes are shown in Table 4 and Table 5. We found that twin pregnancies conceived with ART had a higher rate of asphyxia in the neonate at 1 min and 5 min after delivery (8.7% vs. 4.2%, *p* < 0.05 and 5.3% vs. 2.1%, *p* < 0.05).

Second, we analyzed twin pregnancies complicated by preeclampsia and grouped those women into the with ART group and without ART group. Among the twin pregnancies with preeclampsia, 67 women conceived with ART, and 56 women conceived without ART. We found that twin pregnancies complicated by preeclampsia and conceived with ART had a higher rate of maternal age ≥ 35 (41.8% vs. 16.1%, *p* < 0.05), a higher rate of elderly primipara (35.8% vs. 7.1%, *p* < 0.05), and a lower rate of monochorionic twin pregnancies (11.9% vs. 37.5%, *p* < 0.05) (Table 3). Maternal–neonatal outcome analysis showed that twin pregnancies complicated by preeclampsia and conceived with ART had a higher rate of delivery at gestational week < 34 (29.9% vs. 12.5%, *p* < 0.05), and asphyxia of the neonate at 5 min after delivery (13.4% vs. 1.8%, *p* < 0.05) (Table 4 and Table 5).

Therefore, our study found that ART was a risk factor for twin pregnancies complicated by preeclampsia and led to an increased rate of adverse maternal–neonatal outcomes in twin pregnancies complicated by preeclampsia.

## 4. Discussion

Preeclampsia is a serious complication of twin pregnancy. Many studies have demonstrated that the risk of preeclampsia in pregnant women who conceived with ART was higher than that of pregnant women who conceived naturally [6,20,21]. However, the effect of ART on maternal–neonatal outcomes in twin pregnancies with preeclampsia is unclear. Whether ART treatment increases the risk of adverse pregnancy outcomes in twin pregnancies complicated by preeclampsia warrants further research. Hence, we retrospectively analyzed the epidemiological characteristics and maternal–neonatal outcomes of twin pregnancies and their relationship with ART. Our results show that ART was a risk factor for preeclampsia in twin pregnancies and an increased rate of delivery at gestational week < 34 of the mother and asphyxia of the neonate at 5 min after delivery in twin pregnancies complicated by preeclampsia.

Our study found that the prevalence rate of twin pregnancies with preeclampsia was 17.62% (123/698). In contrast to other studies, our study found a higher prevalence rate. Existing studies have found that the prevalence of preeclampsia in twin pregnancies is between 7.1% and 14.8% [15,18,19]. The analysis of this result may be related to the fact that our hospital accepts not only normal pregnant women but also high-risk pregnant women. Our study found that the prevalence of preeclampsia in twin pregnancies conceived with ART was higher than that in twin pregnancies conceived without ART (Figure 2). Multivariate analysis results showed that ART was a risk factor for preeclampsia in twin pregnancies (AOR: 1.868, 95% CI: 1.187–2.941) after adjusting for maternal factors. The study of Okby et al. (similar to ours) also found that the rate and risk of preeclampsia were higher in the IVF twin pregnancy group than in the spontaneous twin pregnancy group (13.8% vs. 7.6%, *p* < 0.001, OR: 1.81, 95% CI: 1.50–2.17) [15]. Erez et al. analyzed the pregnancy outcomes of twin pregnancies with preeclampsia and normal twin pregnancies and found that IVF was a risk factor for preeclampsia [18]. However, some studies believe that assisted conception is not a risk factor for adverse pregnancy outcomes [22]. This may be related to the fact that the fertility treatment protocols for twin pregnancies in 1990–2003 in the US were different from those in our study in 2013–2021 in China. Additionally, this study excluded instances <24 gestational weeks, where one of the twins was born dead, and of congenital anomalies. These criteria would exclude women with twin pregnancies with severe maternal–neonatal outcomes and would obtain a significantly different result from our study.

Our research found that maternal age ≥ 35 and elderly primipara were higher in twin pregnancies with preeclampsia conceived with ART than in twin pregnancies with preeclampsia conceived without ART. Okby et al. found that in IVF twin pregnancies with preeclampsia, mothers were older and had higher primipara than in twin pregnancies with preeclampsia conceived without ART [15]. Serena et al. found that advanced maternal age (≥40 years) is among the risk factors for twin pregnancies with gestational hypertension/preeclampsia [23]. Our results are similar to those of these studies. This is consistent with the basic characteristics of the ART population; that is, the ART population has multiple factors of infertility, so it is mostly advanced maternal age and primiparous [5]. Studies have shown that advanced age may further increase the risk of adverse maternal–neonatal outcomes [24]. However, after adjusting for maternal factors, our study did not find that maternal age ≥ 35 was a risk factor for preeclampsia. Similar to our result, Lei et al. found that maternal age ≥ 35 years did not increase the risk of preeclampsia when pregnant women were stratified according to maternal age < 35 years and maternal age ≥ 35 years [25]. The reason for this may be related to the fact that advanced age is a population characteristic of ART, which is an obvious difference between the ART and without ART groups, not in the preeclampsia group or the no preeclampsia group.

To further clarify the influence of ART on the pregnancy outcome of twin pregnancies with preeclampsia, we divided the twin pregnancies with preeclampsia into two groups according to whether the mother used ART. Our study found a higher incidence of delivery at gestational weeks < 34 (29.9% vs. 12.5%, *p* < 0.05) and asphyxia of the neonate at 5 min after delivery (13.4% vs. 1.8%, *p* < 0.05) in twin pregnancies with preeclampsia conceived with ART than in the without ART group. Okby et al. analyzed the pregnancy outcomes of twin pregnancies complicated by preeclampsia conceived with IVF and conceived with spontaneous conception, and found that the incidence of early preterm delivery ≤ 34 weeks (23.4% vs. 12.4%, *p* < 0.05) was higher in twin pregnancies complicated by preeclampsia conceived with IVF [15]. However, in the same study, the rate of neonatal Apgar score < 7 at 5 min after delivery was not higher in twin pregnancies with preeclampsia conceived with IVF [15]. The above study mainly analyzed twin pregnancies complicated by preeclampsia conceived with IVF and spontaneously. Our grouping method was similar to that of this study, but the results were different, which may be related to the different ART methods because the time period analyzed in this study (between 1988 and 2010) was earlier than in our study. Sara et al. analyzed the pregnancy outcomes of twin pregnancies conceived with assisted reproductive technology and twin pregnancies conceived with spontaneous conception and found that the risk of gestational age < 28 weeks of the mother and Apgar < 7 of the neonate at 5 min after delivery was higher in the assisted reproductive technology group than in the spontaneous conception group (AOR: 1.3, 95% CI: 1.2–1.4 and AOR: 1.1, 95% CI: 1.0–1.1) after adjusting maternal and paternal age, maternal body mass index, race/ethnicity, smoking status, gestational diabetes, prepregnancy and gestational hypertension, history of preterm delivery and cesarean section [4]. This was similar to our study, but this study did not analyze the influence of preeclampsia on the outcomes of twin pregnancies.

How ART causes adverse pregnancy outcomes in twin pregnancies with preeclampsia may be explained by its effects on utero-placental development and fetal growth. Normal pregnancy requires the coordinated implantation of embryos, trophoblast invasion, and decidualization into the endometrium. However, hormonal abnormalities during the peri-implantation and early placentation period of ART pregnancies lead to the inflammatory pathways and metabolic dysfunction of the endometrium, myometrium, and cervix, including endometrial dysfunction, vascular dysfunction in myometrial arteries, uterine contractility dysfunction, and cervical insufficiency [26]. These processes will reduce the invasion of trophoblasts into the spiral artery of the uterus during remodeling and impair utero-placental development, which can lead to a decrease in blood supply and hypoxia of the placenta and embryo—common in PE. Studies have shown that compared with patients who have low serum levels of estradiol, patients with higher concentrations due to stimulation during ART could have a higher risk of preeclampsia [27]. The increase in estradiol levels that occurs in late pregnancy prevents further remodeling, eventually causing preeclampsia [28]. A twin pregnancy alone increases the risk of hypertensive disorders and preeclampsia [29]. The reason for this may be associated with the fact that the weight, volume, and relative ishemia of the placenta are higher in twin pregnancies than in singleton pregnancies [30]. In addition, studies have confirmed that IVF treatment interferes with the epigenetic mechanism that regulates fetal growth and development, leading to adverse maternal and child outcomes [31,32].

Studies have shown that infertility itself, rather than ART, increases the risk of adverse outcomes [33]. Infertility and reproductive disorders are the reasons that couples use ART, and these disorders can include endometriosis, adenomyosis, PCOS, and abnormal immune system. Each of these disorders may concomitantly increase the risk of hypertensive disorders and adverse maternal–neonatal outcomes via inflammatory immune reactions, abnormal hormone levels, placenta spiral vessel abnormalities, and decidual senescence mechanisms [5,26]. Studies have shown that the risk of perinatal death of the fetus is higher in infertile women without the treatment of ART, than in fertile women. Therefore, infertility itself may be the cause of the adverse outcome [34]. Compared with women who conceived spontaneously, patients with infertility or reduced fertility had an increased rate and risk of preeclampsia after pregnancy [25,35,36]. PCOS is among the causes of infertility. Studies have shown that PCOS may be strongly related to pregnancy-induced hypertension and/or preeclampsia [37]. However, our study did not find a significant difference in PCOS between the ART twin pregnancies with preeclampsia group and the twin pregnancies with preeclampsia conceived without ART group. The main reasons for the different results are the different grouping methods and research purposes.

Monochorionic–monoamniotic twin pregnancies lead to a higher risk of adverse maternal–neonatal outcomes [38]. Our study found that not only ART but also monochorionic twin pregnancies increase the risk of preeclampsia. In accordance with our study, Karien et al. found that the assisted conception group in monochorionic twins had a higher rate of preeclampsia than the spontaneous conception group, although the difference was not statistically significant [39]. Another study found that the rate of monochorionic twin pregnancies with preeclampsia was not higher than that in twin pregnancies without preeclampsia, which may be related to the exclusion of monochorionic–monoamniotic twin pregnancies in this study [19].

Our study retrospectively analyzed the effect of ART on pregnancy outcomes in twin pregnancies with preeclampsia. Since the ACOG guidelines about hypertension in pregnancy in 2013 no longer considered proteinuria as a prerequisite for the diagnosis of preeclampsia, which would lead to a different diagnosis of preeclampsia before 2013 and after 2013, this study was based on the ACOG guidelines published in 2013 and included 8 years of twin pregnancy data for retrospective analysis [10]. In addition, to better improve maternal–neonatal health for worldwide health care policy, our study supplemented twin pregnancy data in Tianjin, China, in recent years, which is helpful to provide clinical data for the management of twin pregnancies worldwide. However, our research has some limitations. Our research was retrospective in design and was not based on medical records for the purpose of our study, and some data were, therefore, missed, such as body weight. Then, we used PCOS as a substitution, which is known to be characteristic of obesity in PCOS patients, and analyzed the effect of PCOS on preeclampsia in twin pregnancies. We did not obtain data on the cause of infertility, duration of infertility, protocols used for ART treatment, the number of embryos transferred, or specific methods used for embryo preservation. Therefore, we did not analyze the effects of different assisted conception methods on pregnancy outcomes. Our study found that ART led to an increased rate of adverse maternal–neonatal outcomes in twin pregnancies complicated by preeclampsia. In the future, transferring relatively few and high-quality embryos at a time will greatly reduce the twin pregnancy rate and adverse pregnancy outcome rate [40,41]. Furthermore, selective fetal reduction is not a good way to decrease the complications of twin pregnancy [42,43,44]. Hence, elective single embryo transfer is a feasible approach to avoid additional complications.

## 5. Conclusions

Our research suggested that twin pregnancies conceived with ART had a higher risk of preeclampsia than twin pregnancies conceived without ART. Twin pregnancies complicated by preeclampsia and conceived by ART had a higher rate of delivery at gestational week < 34 and asphyxia of the neonate at 5 min after delivery than twin pregnancies complicated by preeclampsia and conceived spontaneously. One efficient approach is to transfer an elective single embryo and freeze the remaining ones to reduce the risk of twin pregnancies and preeclampsia and to improve maternal–neonatal outcomes.

## Figures and Tables

**Figure 1 diagnostics-12-01334-f001:**
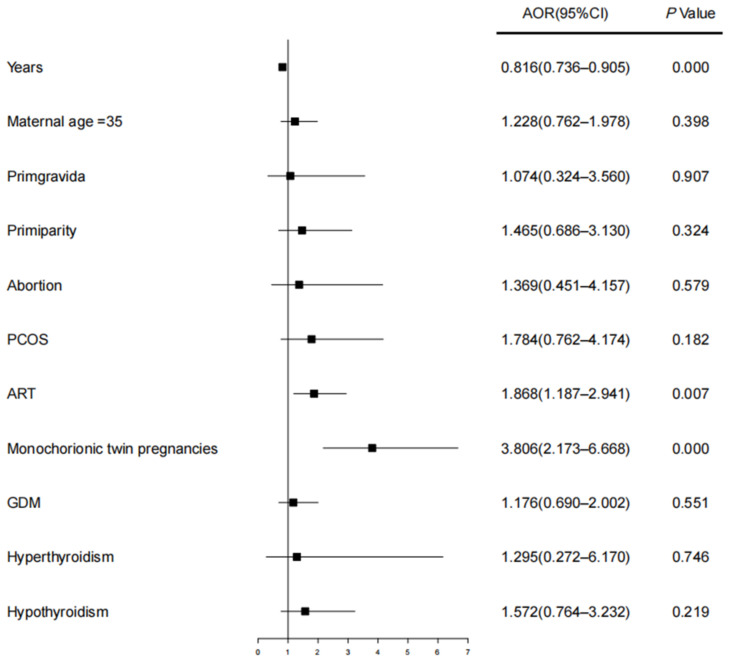
Forest plot showing risk factors for preeclampsia in twin pregnancies. Abbreviations: AOR, adjusted odds ratio; 95% CI, 95% confidence intervals; ART, assisted reproductive technology; PCOS, polycystic ovary syndrome; GDM, gestational diabetes mellitus. We adjusted for years, maternal age, gravidity, parity, abortion, PCOS, chorionicity, GDM, hyperthyroidism, and hypothyroidism.

**Figure 2 diagnostics-12-01334-f002:**
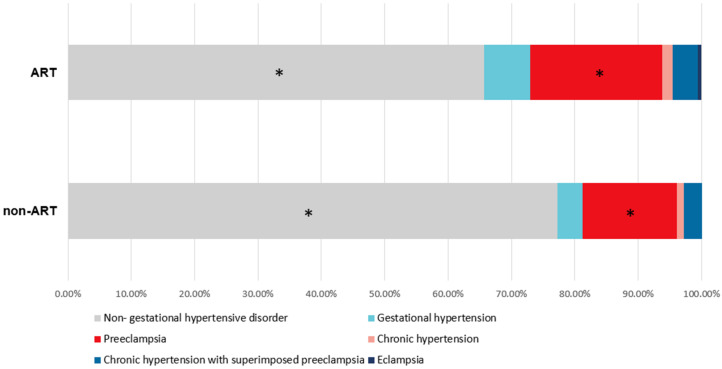
Hypertension disorders in ART and non-ART group. * *p* < 0.05 was considered significant.

**Table 1 diagnostics-12-01334-t001:** Prevalence of twin pregnancies with hypertensive disorders.

	No.	%
Twin pregnancies	698	
Gestational hypertension	38	5.44
Preeclampsia	123	17.62
Chronic hypertension	9	1.29
Chronic hypertension with superimposed preeclampsia	24	3.44
Eclampsia	2	0.29

**Table 2 diagnostics-12-01334-t002:** Risk factors for preeclampsia in twin pregnancies.

	OR (95% CI)	*p*	AOR (95% CI)	*p*
Years	0.830 (0.750–0.919)	0.000	0.816 (0.736–0.905)	0.000
Maternal age ≥ 35	1.241 (0.809–1.905)	0.323	1.228 (0.762–1.978)	0.398
Primigravida	0.891 (0.604–1.316)	0.563	1.074 (0.324–3.560)	0.907
Primiparity	1.329 (0.763–2.315)	0.316	1.465 (0.686–3.130)	0.324
Abortion	1.200 (0.810–1.778)	0.364	1.369 (0.451–4.157)	0.579
PCOS	2.083 (0.930–4.665)	0.075	1.784 (0.762–4.174)	0.182
ART	1.512 (1.023–2.236)	0.038	1.868 (1.187–2.941)	0.007
Monochorionic twin pregnancies	2.804 (1.704–4.614)	0.000	3.806 (2.173–6.668)	0.000
GDM	1.119 (0.677–1.851)	0.660	1.176 (0.690–2.002)	0.551
Hyperthyroidism	0.934 (0.202–4.316)	0.930	1.295 (0.272–6.170)	0.746
Hypothyroidism	1.572 (0.794–3.110)	0.194	1.572 (0.764–3.232)	0.219

Abbreviations: OR, odds ratio; AOR, adjusted odds ratio; 95% CI, 95% confidence intervals; ART, assisted reproductive technology; PCOS, polycystic ovary syndrome; GDM, gestational diabetes mellitus. We adjusted for years, maternal age, gravidity, parity, abortion, PCOS, chorionicity, GDM, hyperthyroidism, and hypothyroidism.

**Table 3 diagnostics-12-01334-t003:** Clinical characteristics of twin pregnancies with preeclampsia and their association with ART.

	Twin Pregnancies		Twin Pregnancies with Preeclampsia			
	ART (*n* = 321)	Non-ART (*n* = 377)	ART (*n* = 67)	Non-ART (*n* = 56)	*p* ^#^	** *p* ^&^ **
	NO. (%)	NO. (%)	NO. (%)	NO. (%)		
Maternal age (Mean ± SD)	33.31 ± 4.365	30.83 ± 4.337	33.97 ± 5.483	30.23 ± 4.943	0.000 ^a^	0.000 ^a^
Maternal age ≥ 35	123 (38.3%)	62 (16.4%)	28 (41.8%)	9 (16.1%)	0.000 ^b^	0.002 ^b^
Inpatient days (Mean ± SD)	6.80 ± 3.925	6.09 ± 3.273	8.12 ± 3.867	7.00 ± 3.506	0.010 ^a^	0.098 ^a^
Gravidity					0.879 ^b^	0.230 ^b^
Primigravida	173 (53.9%)	201 (53.3%)	31 (46.3%)	32 (57.1%)		
Multigravida	148 (46.1%)	176 (46.7%)	36 (53.7%)	24 (42.9%)		
Parity (*n*, %)					0.000 ^a^	0.509 ^b^
Primiparity	288 (89.7%)	292 (77.5%)	59 (88.1%)	47 (83.9%)		
Multiparity	33 (10.3%)	85 (22.5%)	8 (11.9%)	9 (16.1%)		
Abortion	136 (42.4%)	145 (38.5%)	33 (49.3%)	21 (37.5%)	0.294 ^b^	0.191 ^b^
Elderly primipara	105 (32.7%)	32 (8.5%)	24 (35.8%)	4 (7.1%)	0.000 ^b^	0.000 ^b^
PCOS	19 (5.9%)	11 (2.9%)	6 (9.0%)	3 (5.4%)	0.051 ^b^	0.678 ^c^
Monochorionic twin pregnancies	14 (4.4%)	72 (19.1%)	8 (11.9%)	21 (37.5%)	0.000 ^b^	0.001 ^b^
GDM	67 (20.9%)	54 (14.3%)	14 (20.9%)	9 (16.1%)	0.023 ^b^	0.494 ^b^
Hyperthyroidism	5 (1.6%)	7 (1.9%)	1 (1.5%)	1 (1.8%)	0.726 ^b^	1.000 ^d^
Hypothyroidism	29 (9.0%)	20 (5.3%)	6 (9.0%)	6 (10.7%)	0.055 ^b^	0.743 ^b^

^#^ Twin pregnancies conceived with ART vs. twin pregnancies conceived without ART; ^&^ twin pregnancies with preeclampsia conceived with ART vs. twin pregnancies with preeclampsia conceived without ART; ^a^ Student’s *t*-test; ^b^ Pearson’s Chi-square test; ^c^ continuity correction; ^d^ Fisher’s exact test. Abbreviations: SD, standard deviation; ART, assisted reproductive technology; PCOS, polycystic ovary syndrome; GDM, gestational diabetes mellitus.

**Table 4 diagnostics-12-01334-t004:** Maternal outcomes of twin pregnancies with preeclampsia and their association with ART.

	Twin Pregnancies		Twin Pregnancies with Preeclampsia			
	ART (*n* = 321)	Non-ART (*n* = 377)	ART (*n* = 67)	Non-ART (*n* = 56)	*p* ^#^	*p* ^&^
	NO. (%)	NO. (%)	NO. (%)	NO. (%)		
Mode of delivery						
Labor	15 (4.7%)	29 (7.7%)	0	0	0.102 ^c^	-
Caesarean section	306 (95.3%)	348 (92.3%)	60 (100%)	56 (100%)		
Delivery weeks (Mean ± SD)	36.833 ± 22.152	35.720 ± 3.171	34.744 ± 2.918	35.541 ± 2.962	0.373 ^a^	0.137 ^b^
ICU	19 (5.9%)	16 (4.2%)	12 (17.9%)	12 (21.4%)	0.312 ^c^	0.624 ^c^
Heart failure	8 (2.5%)	8 (2.1%)	6 (9.0%)	7 (12.5%)	0.745 ^c^	0.524 ^c^
Delivery gestational week < 37	170 (53.0%)	184 (48.8%)	50 (74.6%)	37 (66.1%)	0.274 ^c^	0.299 ^c^
Delivery gestational week < 34	64 (19.9%)	59 (15.6%)	20 (29.9%)	7 (12.5%)	0.138 ^c^	0.021 ^c^
Delivery gestational week < 32	36 (11.2%)	39 (10.3%)	11 (16.4%)	4 (7.1%)	0.711 ^c^	0.117 ^c^
Delivery gestational week < 28	9 (2.8%)	13 (3.4%)	3 (4.5%)	2 (3.6%)	0.627 ^c^	1.000 ^d^
HELLP	6 (1.9%)	3 (0.8%)	5 (7.5%)	3 (5.4%)	0.360 ^d^	0.917 ^d^
PROM	61 (19.0%)	71 (18.8%)	9 (13.4%)	5 (8.9%)	0.954 ^c^	0.433 ^c^
PPROM	52 (16.2%)	55 (14.6%)	9 (13.4%)	2 (3.6%)	0.556 ^c^	0.056 ^c^
Placenta previa	10 (3.1%)	9 (2.4%)	0	3 (5.4%)	0.556 ^c^	0.183 ^d^
Placenta accreta	8 (2.5%)	2 (0.5%)	5 (7.5%)	1 (1.8%)	0.064 ^d^	0.301 ^d^
Placental abruption	5 (1.6%)	5 (1.3%)	1 (1.5%)	1 (1.8%)	1.000 ^d^	1.000 ^e^
Postpartum hemorrhage	20 (6.2%)	22 (5.8%)	4 (6.0%)	4 (7.1%)	0.827 ^c^	1.000 ^d^
Twin to twin transfusion syndrome	0	6 (1.6%)	0	2 (3.6%)	0.063 ^d^	0.205 ^e^

^#^ Twin pregnancies conceived with ART vs. twin pregnancies conceived without ART; ^&^ twin pregnancies with preeclampsia conceived with ART vs. twin pregnancies with preeclampsia conceived without ART; ^a^ Student’s *t*-test; ^b^ Wilcoxon rank-sum test; ^c^ Pearson’s Chi-square test; ^d^ continuity correction; ^e^ Fisher’s exact test. Abbreviations: ART, assisted reproductive technology; SD, standard deviation; ICU, intensive care unit; HELLP syndrome, hemolysis, elevated liver enzymes, and low platelet count syndrome.

**Table 5 diagnostics-12-01334-t005:** The neonatal outcomes of twin pregnancies with preeclampsia and their association with ART.

	Twin Pregnancies		Twin Pregnancies with Preeclampsia			
	ART (*n* = 321)	Non-ART (*n* = 377)	ART (*n* = 67)	Non-ART (*n* = 56)	*p* ^#^	*p* ^&^
	NO. (%)	NO. (%)	NO. (%)	NO. (%)		
NICU	158 (49.2%)	168 (44.6%)	40 (59.7%)	33 (58.9%)	0.219 ^a^	0.931 ^a^
Low birth weight	188 (58.6%)	232 (61.5%)	48 (71.6%)	37 (66.1%)	0.424 ^a^	0.506 ^a^
Very low birth weight	35 (10.9%)	43 (11.4%)	13 (19.4%)	7 (12.5%)	0.834 ^a^	0.302 ^a^
Asphyxia at 1 min	28 (8.7%)	16 (4.2%)	11 (16.4%)	5 (8.9%)	0.015 ^a^	0.219 ^a^
Asphyxia at 5 min	17 (5.3%)	8 (2.1%)	9 (13.4%)	1 (1.8%)	0.025 ^a^	0.043 ^b^
Neonatal malformation	13 (4.0%)	10 (2.7%)	6 (9.0%)	3 (5.4%)	0.303 ^a^	0.678 ^b^
Perinatal mortality	17 (5.3%)	17 (4.5%)	4 (6.0%)	2 (3.6%)	0.630 ^a^	0.846 ^b^

^#^ Twin pregnancies conceived with ART vs. twin pregnancies conceived without ART; ^&^ twin pregnancies with preeclampsia conceived with ART vs. twin pregnancies with preeclampsia conceived without ART; ^a^ Pearson’s Chi-square test; ^b^ continuity correction. Abbreviations: ART, Assisted Reproductive Technology; NICU, neonatal intensive care unit.

## Data Availability

The data presented in this study are available on request from the corresponding author.

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
