# Peer review of "The Maternal–Neonatal Outcomes of Twin Pregnancies with Preeclampsia and Their Association with Assisted Reproductive Technology: A Retrospective Study"

_diagnostics, 2022, doi:10.3390/diagnostics12061334_

Round 1

Reviewer 1 Report

This paper describes a retrospective study in which the investigators analyzed twin pregnancies to determine the rate of preeclampsia and the influence of ART on these rates and on pregnancy outcomes. The study population included 709 twin pregnancies over an 8 1/2-year period from a single hospital. ART pregnancies were compared with spontaneous pregnancies (i.e., non-ART pregnancies). The rate of preeclampsia was higher in the ART than in the spontaneous pregnancies and twin pregnancies with preeclampsia conceived with ART had a higher rate of several delivery complications including preterm premature rupture of membranes and neonatal asphyxia. They conclude that ART increases the risk of preeclampsia and of adverse outcomes compared with non-ART conception.

The paper presents interesting data but the justification for studying twins only is not given. The authors should provide a reason as to why twin pregnancies and not all pregnancies were studied.

The term SP is confusing and misused. SP (which is not defined in the Abstract) is defined in the Introduction as Spontaneous Pregnancy. The authors then go on to use the term in sentences that say things such as "those conceived with SP". Saying "those conceived with spontaneous pregnancy" doesn't mean anything.  SP should be removed and the authors should say "those conceived without ART. In addition. ART needs to be clearly defined. Is this IVF and related procedures only or is this IVF, OI, IUI and so forth? If the former, then use of ART in one location and IVF in another is very confusing and consistent use of ART only would be better. Further, if ART includes only IVF techniques then saying that all others are spontaneous pregnancies is not correct since other procedures may have been used. It would be better to list these as ART and non-ART conceptions.

In the second paragraph of the Methods section, the authors state that the "women were time matched". What does this mean? From other information in Methods it appears that all twin pregnancies were studied so what sort of matching was done? If "time matched" what time is this? If matched, why not match for age, parity, prior preeclampsia, and other relevant characteristics? The paragraph on strengths in the Discussion also lists time matching as a strength. This sentence should be removed unless a clear description of matching is given.

We are never told whether there is only one delivery per woman included in study sample. If more than one, what statistical methodology was used to adjust for more than one delivery per woman?

It does not appear that models were adjusted for year of delivery. Year should be added to the models since the time span from 2014-2021 included a period of major changes in ART methodology.

Minor points:

The Abstract mentions that the study included "21,922 women with twin pregnancies". This is misleading. The study included 709 twin pregnancies. The larger number is total pregnancies delivered at the hospital but other than providing a denominator for twin rate the majority of these were not studied. The number 21,922 should be removed from the Abstract.

Other than saying this was a retrospective study, the Abstract says nothing about methodology. More information on methods should be added, if possible within the word limit.

If this was a retrospective study, how did all participants provide informed consent? Also, to whom would they have provided consent and at what point in their treatment would consent have been provided?

In calculating gestational age, how was LMP determined for ART pregnancies? Was it calculated on the basis of day of transfer or was only crown rump length used for the ART pregnancies?

Chronic hypertension is listed in the Methods as an outcome. How is this an outcome given that it occurs before the intervention (ART or non-ART)? It is actually an underlying condition. The same is the case for hypo and hyper thyroidism unless these changed following the pregnancy.

"Fingding" in the next to last paragraph of Methods should be changed to "finding". The sentence is also missing the word "of"...."ultrasound finding of the placenta...."

Abortion is presented in Table 1 and adjusted for in the models. Please define abortion. Does it include spontaneous abortion (miscarriage) as well as therapeutic abortion?

Table 2 could be simplified by removing the "no" lines and presenting "yes" only.  

Table 3 needs a footnote with what was adjusted for in the models.

The first line of the Discussion doesn't make sense. What does "great guiding significance for clinical diagnosis and treatment" mean?

The word "proven" should be removed from the second sentence of the Discussion and replaced with "demonstrated".

In paragraph 2 of the Discussion, remove "such as" from the list of items adjusted for in the models. The full list is given.

The last sentence of paragraph 2 of the discussion needs further explanation. What is meant by "population and the fertility treatment protocol were different from the current treatment protocol, and grouping principles were also different"? Please provide some specifics.

In citing reference 16 in the Discussion the authors refer to the paper by OFFER et al. This is incorrect. The first author's name is Erez.

In paragraph 6 of the Discussion (beginning with "It is not yet clear how ART causes..."), the authors present information as though it is well established when virtually everything they are saying is only hypothetical. The wording should be changed in this paragraph to make the hypothetical nature of the discussion clear.

In paragraph 7 of the Discussion the authors cite reference #31 to say that "infertile women who are not treated with ART have an increased risk of perinatal death.." It is not the women who have the increased risk of death after delivery, it is infants. This should be clarified by rewording the sentence.

What is meant in the Discussion by "...different regions have different medical levels.."?

Author Response

Dear reviewer,

We are sincerely grateful for your insightful and constructive comments. Your comments are much beneficial to improving the quality and readability of our paper. We have addressed all the comments carefully, and the revised portions are marked up using the “Track Changes” function in the manuscript. And we have uploaded the revised version of the manuscript as an attachment. Revision notes, point-to-point, are given as follows:

Point 1: The paper presents interesting data but the justification for studying twins only is not given. The authors should provide a reason as to why twin pregnancies and not all pregnancies were studied.

Response 1: Thank you for your meaningful suggestion. Modifying this content according to your suggestion will improve the overall understanding of the manuscript. In the Introduction, we explained the reasons for our focusing on twin pregnancy rather than the whole pregnant population. If you have any suggestions for modification, please do not hesitate to correct us.

Point 2: The term SP is confusing and misused. SP (which is not defined in the Abstract) is defined in the Introduction as Spontaneous Pregnancy. The authors then go on to use the term in sentences that say things such as "those conceived with SP". Saying "those conceived with spontaneous pregnancy" doesn't mean anything.  SP should be removed and the authors should say "those conceived without ART. In addition. ART needs to be clearly defined. Is this IVF and related procedures only or is this IVF, OI, IUI and so forth? If the former, then use of ART in one location and IVF in another is very confusing and consistent use of ART only would be better. Further, if ART includes only IVF techniques then saying that all others are spontaneous pregnancies is not correct since other procedures may have been used. It would be better to list these as ART and non-ART conceptions.

Response 2: Thank you for the very meaningful advice. It is very helpful for us to express more accurately. We replaced “SP” with “conceived spontaneously” in the whole paper and “non-ART “ in the table and figure. We reviewed the medical records and defined the ART involved in our study. ART here mainly refers to in vitro fertilization and embryo transfer techniques. If you think a further correction is needed, please do not hesitate to correct us.

Point 3: In the second paragraph of the Methods section, the authors state that the "women were time matched". What does this mean? From other information in Methods it appears that all twin pregnancies were studied so what sort of matching was done? If "time matched" what time is this? If matched, why not match for age, parity, prior preeclampsia, and other relevant characteristics? The paragraph on strengths in the Discussion also lists time matching as a strength. This sentence should be removed unless a clear description of matching is given.

Response 3: Thank you for your thoughtful comments. We deleted “time-matched” after careful thinking. What we wanted to express originally is that these women with twin pregnancies delivered in the period of  December 2003 to September 2021, but time matched was not appropriate here. We have accepted your suggestion with gratitude.

Point 4: We are never told whether there is only one delivery per woman included in study sample. If more than one, what statistical methodology was used to adjust for more than one delivery per woman?

Response 4: Thank you for your significant reminding. We have checked the data again and found that some patients had duplicate names but no duplicate clinical data. In addition, due to the restrictions of China's fertility policy, it is less likely to have multiple children or twin babies in the second pregnancy. We confirmed that each patient delivered only once. You can find this in the second paragraph of Materials and Methods. 

Point 5: It does not appear that models were adjusted for year of delivery. Year should be added to the models since the time span from 2014-2021 included a period of major changes in ART methodology.

Response 5: Thank you for your valuable counsel. We have checked the data for years and adjusted it in our models. Please see this in table 2 and figure 1.

Minor points:

Point 1: The Abstract mentions that the study included "21,922 women with twin pregnancies". This is misleading. The study included 709 twin pregnancies. The larger number is total pregnancies delivered at the hospital but other than providing a denominator for twin rate the majority of these were not studied. The number 21,922 should be removed from the Abstract.

Response 1: Thank you so much for pointing this out. We apologize for our negligence that led to misexpression. We have corrected it in the Abstract.

Point 2: Other than saying this was a retrospective study, the Abstract says nothing about methodology. More information on methods should be added, if possible within the word limit.

Response 2: Thanks a lot for your thoughtful comment. We have added the description of the methodology in the Abstract. Please see Methods of Abstract.

Point 3: If this was a retrospective study, how did all participants provide informed consent? Also, to whom would they have provided consent and at what point in their treatment would consent have been provided?

Response 3: Thanks for your valuable suggestion. We apologize for the confusion. We put it regarding other people's research when writing this article. And now we have realized that this sentence is not suitable for our research. We have deleted it after careful consideration of your comments.

Point 4: In calculating gestational age, how was LMP determined for ART pregnancies? Was it calculated on the basis of day of transfer or was only crown rump length used for the ART pregnancies?

Response 4: Thanks so much for your valuable advice. We apologize for our negligence, the description here is inaccurate, and we have corrected it. Because the transplantation date and implantation date of IVFET pregnancy are different from those conceived spontaneously, it is not appropriate to calculate the due date according to the date of the last menstruation. We have corrected the method for calculating gestational age in ART pregnancies as the crown-rump lengths of the fetuses by Doppler ultrasonography as we did in clinical practice. At present, we indeed used different methods to calculate gestational weeks in different populations in our hospital. Please see this in the fourth paragraph in Methods.

Point 5: Chronic hypertension is listed in the Methods as an outcome. How is this an outcome given that it occurs before the intervention (ART or non-ART)? It is actually an underlying condition. The same is the case for hypo and hyper thyroidism unless these changed following the pregnancy.

Response 5: Thanks for your suggestion. Chronic hypertension, chronic hypertension with preeclampsia, and preeclampsia are different diagnoses. In this study, patients with preeclampsia were not included in patients with chronic hypertension with preeclampsia, so the effects of chronic hypertension and chronic hypertension on preeclampsia could not be analyzed.

Point 6: "Fingding" in the next to last paragraph of Methods should be changed to "finding". The sentence is also missing the word "of"...."ultrasound finding of the placenta...."

Response 6: Thanks a lot for your reminding. We are very sorry for our incorrect writing and it is rectified at the fifteenth line in the seventh paragraph of Materials and Methods.

Point 7: Abortion is presented in Table 1 and adjusted for in the models. Please define abortion. Does it include spontaneous abortion (miscarriage) as well as therapeutic abortion?

Response 7: Your suggestion is meaningful to make the manuscript more correct. We have defined abortion in the seventh paragraph of Materials and Methods.

Point 8: Table 2 could be simplified by removing the "no" lines and presenting "yes" only.  

Response 8: Your suggestion is very useful to us. We have removed the "no" lines according to your suggestions. You can find rectifications in all Tables. 

Point 9: Table 3 needs a footnote with what was adjusted for in the models.

Response 9: Thanks for your thoughtful counsel. We have added footnotes. Please see the current Table 2 and figure 1. 

Point 10: The first line of the Discussion doesn't make sense. What does "great guiding significance for clinical diagnosis and treatment" mean?

Response 10: Your suggestion can make the text clear, and we have deleted it. Please see the first line of the Discussion.

Point 11: The word "proven" should be removed from the second sentence of the Discussion and replaced with "demonstrated".

Response 11: Your suggestion is very important to accurately express the content of the manuscript. Thank you for your suggestion and we have rectified it. Please see the second sentence of the Discussion.

Point 12: In paragraph 2 of the Discussion, remove "such as" from the list of items adjusted for in the models. The full list is given.

Response 12: Thank you for your valuable suggestion. Your suggestion is meaningful to make the manuscript more correct. We have removed it. Please see the second paragraph of the Discussion.

Point 13: The last sentence of paragraph 2 of the discussion needs further explanation. What is meant by "population and the fertility treatment protocol were different from the current treatment protocol, and grouping principles were also different"? Please provide some specifics.

Response 13: Thanks a lot for your constructive comment. We apologize for the confusion caused to you. We have rewritten this part. Please see the last part of the second paragraph of the Discussion.

Point 14: In citing reference 16 in the Discussion the authors refer to the paper by OFFER et al. This is incorrect. The first author's name is Erez.

Response 14: Thank you for pointing this out. We apologize for our negligence and we have corrected it. Please see paragraph 2 in the Discussion.

Point 15: In paragraph 6 of the Discussion (beginning with "It is not yet clear how ART causes..."), the authors present information as though it is well established when virtually everything they are saying is only hypothetical. The wording should be changed in this paragraph to make the hypothetical nature of the discussion clear.

Response 15: Thank you for your valuable counsel. The revision made the manuscript more reasonable. We have revised it. Please paragraph 5 of the Discussion.

Point 16: In paragraph 7 of the Discussion the authors cite reference #31 to say that "infertile women who are not treated with ART have an increased risk of perinatal death.." It is not the women who have the increased risk of death after delivery, it is infants. This should be clarified by rewording the sentence.

Response 16: Thank you for your significant reminding. We apologize for the incorrect expression and we have corrected it. Please see paragraph 6 of the Discussion.

Point 17: What is meant in the Discussion by "...different regions have different medical levels.."?

Response 17: Thanks for your excellent advice. We apologize for the confusion caused to you. We have rewritten this part with some revisions. Please see this in the last paragraph of the Discussion.

Thank you again for your positive and constructive comments and suggestions on our manuscript. If you think any further correction is needed, please do not hesitate to contact us.

We hope you will find our revised manuscript acceptable for publication.

Sincerely,

Dr. Li

Reviewer 2 Report

Thank for you give to me an opportunity to review an article as “The Maternal-neonatal Outcomes of Twin Pregnancies with Preeclampsia and their Association with Assisted Reproductive Technology: A Retrospective Study.

Preeclampsia is one of the five leading causes of death in pregnant women. It is a fatal disease that causes fetal growth failure or sudden death of the fetus if it leads to eclampsia, which occurs when seizures occur. Therefore, ART could increase the reason for twin pregnancies and increase pre-eclampsia.

Based on the information, this article provides a valuable study of ART and the pre-eclampsia relationship.

However, I have some comments and suggest adding more data to improve the quality of the manuscript for publication.

First. The author emphasized that ART leads to twin pregnancy compared with spontaneous pregnancy.

In the introduction part. There is a reference regarding ART improving twin pregnancy. But there is no reference that ART itself induces the fatal defect. 

I strongly suggest that need to add more factors like the number of embryos and ART patient age to the table. And please make a plot of preeclampsia ratios of SP vs ART.  A plot is more visible and easier to understand for compared data.

Author Response

Dear reviewer,

Thank you very much for your kindly comments on our manuscript. There is no doubt that these comments are valuable and very helpful for revising and improving our manuscript. In what follows, we would like to answer the questions you mentioned and give a detailed account of the changes made to the original manuscript.

Point 1: First. The author emphasized that ART leads to twin pregnancy compared with spontaneous pregnancy.

In the introduction part. There is a reference regarding ART improving twin pregnancy. But there is no reference that ART itself induces the fatal defect.

Response 1: Thank you for your valuable counsel. We have added relevant references. Please see the Introduction. If you think there are still areas that need to be modified, please feel free to contact us.

Point 2: I strongly suggest that need to add more factors like the number of embryos and ART patient age to the table. And please make a plot of preeclampsia ratios of SP vs ART. A plot is more visible and easier to understand for compared data.

Response 2: Thank you for your meaningful advice. As this is a retrospective study, no detailed description of the number of transferred embryos was found after reviewing the medical records. In addition, most of the patients in the early years cannot be contacted by the phone number they kept in our hospital and the information could not be tracked. Therefore, the factor of the number of embryos could not be added to the table. Please see the last paragraph of the Discussion. The ages of ART patients are shown in Tables 2 and Tables 3 and analyzed in the regression analysis of Table 2. Table 2 is a plot of preeclampsia ratios of SP vs ART. If you think a further correction is needed, please do not hesitate to contact us.

Thank you again for your positive and constructive comments and suggestions on our manuscript.

We hope you will find our revised manuscript acceptable for publication.

Sincerely,

Dr. Li

Reviewer 3 Report

This is a retrospective study that aimed to investigate the maternal-neonatal outcomes of twin pregnancies with preeclampsia and their association with ART. Authors concluded that  ART increases the risk of preeclampsia for twin pregnancies and the rate of adverse maternal-neonatal outcomes for twin pregnancies with preeclampsia.

The cohort of participants that authors have used is big; they have applied plenty of analysis and their methodology in general is acceptable. But, there are some ponts that need reconsideration and improvement.

The rationale of the study is not clear: authors should reconsider.

Mild amendments in the full paper in syntax and language would improve the quality of the paper.

The parameter of egg donation has included high risk selection biases. A regression analysis or subgroup analysis might be necessary, or the exclusion of this group.

There is a need for sample size calculation as the comparisons are plenty. This would give with high accuracy the power of this study.

Another point of the analysis is that there were differences in the study population form the stage of their demographic characteristics; thus, the final results is impaired; to deal with this, authors should perform regression analysis for those parameters (which partially they did) and conclude in relation to those parameters. The final conclusion should be based also to the quality of the study.

The reference list should be expanded.

Finally, I am not sure if this particular paper suits to the specific section “point of care diagnostics and devices”.

Author Response

Dear reviewer,

Thank you very much for the time and effort you spent reviewing our manuscript, as well as the valuable and constructive suggestions. There is no doubt that these suggestions are very helpful for revising and improving our manuscript. In what follows, we would like to answer the questions you mentioned and give a detailed account of the changes made to the original manuscript. And we have uploaded the revised version of the manuscript as an attachment.

Point 1: The rationale of the study is not clear: authors should reconsider.

Response 1: Thank you for your suggestion. Your suggestion will benefit the correctness and comprehensibility of the manuscript and bring the manuscript closer to the requirements for publishing. We have revised it according to your opinion. Please see the Abstract and Conclusion.

Point 2: Mild amendments in the full paper in syntax and language would improve the quality of the paper.

Response 2: Thank you for your valuable advice. According to your advice, this manuscript was edited for proper English language, grammar, punctuation, spelling, and overall style by editors at MDPI Language Editing Services. If you think it needs to be revised, please do not hesitate to correct me.

Point 3: The parameter of egg donation has included high risk selection biases. A regression analysis or subgroup analysis might be necessary, or the exclusion of this group.

Response 3: Thanks for your valuable comments. We reviewed the medical records, only one patient with twin pregnancy underwent egg donation, and we deleted her data. Please see the description of ART in the Materials and Methods.

Point 4: There is a need for sample size calculation as the comparisons are plenty. This would give with high accuracy the power of this study.

Response 4: Regarding the sample size calculation, we think that your suggestion is very meaningful. We summarized the calculation formula of sample size according to previous literature and calculated the sample size of our study. Please see paragraph 8 in the Materials and Methods. If you think that it needs to be revised, please do not hesitate to contact us.

Point 5: Another point of the analysis is that there were differences in the study population form the stage of their demographic characteristics; thus, the final results is impaired; to deal with this, authors should perform regression analysis for those parameters (which partially they did) and conclude in relation to those parameters. The final conclusion should be based also to the quality of the study.

Response 5: Thank you for your valuable advice. Regarding the stage of twin pregnancy demographic characteristics, we believe that your comment is very meaningful. To ensure the consistency of demographic characteristics, we carried out a regression analysis of the year in order to improve the research quality. Please see Table2 and Figure1. If you have any suggestions for modification, please do not hesitate to contact us.

Point 6: The reference list should be expanded.

Response 6: Thanks a lot for your constructive comment. We added new references. If you think that it’s not enough, please do not hesitate to contact us.

Point7: Finally, I am not sure if this particular paper suits to the specific section “point of care diagnostics and devices”.

Response 7: Thanks for your thoughtful suggestion. We understand the core content of the specific section “point of care diagnostics and devices”. And we have made an effort to make corrections to suit the content of this section. If you think a further correction is needed, please do not hesitate to contact us.

Thank you again for your positive and constructive comments and suggestions on our manuscript. If you think any further correction is needed, please do not hesitate to contact us.

We hope you will find our revised manuscript acceptable for publication.

Sincerely,

Dr. Li

Reviewer 4 Report

The paper is typical of an analysis of a clinical data report in relation
to a specific community. The work should end with an unequivocal statement that
the only way to avoid many complications (not only hypertension) of
multiplefetus pregnancy is to transfer one embryo and freeze the remaining ones.

Author Response

Dear reviewer,

Thank you very much for the time and effort you spent reviewing our manuscript, we appreciate your kindly decision and constructive comments on our manuscript. There is no doubt that these suggestions are very helpful for revising and improving our manuscript. We have revised our manuscript according to your valuable comments. And we have uploaded the revised version of the manuscript as an attachment. The following is our response to your comments.

Point: The paper is typical of an analysis of a clinical data report in relation to a specific community. The work should end with an unequivocal statement that the only way to avoid many complications (not only hypertension) of multiplefetus pregnancy is to transfer one embryo and freeze the remaining ones.

Response: Thank you for your valuable counsel. Regarding the unequivocal statement, we think that your suggestion is very meaningful. We have supplemented it in the Abstract, Discussion, and Conclusion. If you think it needs to be revised, please do not hesitate to correct us.

Thank you again for your positive and constructive comments and suggestions on our manuscript.

We hope you will find our revised manuscript acceptable for publication.

Sincerely,

Dr. Li

Round 2

Reviewer 3 Report

I think most issues have been addressed. I only would like to suggest authors to take a second look at the suggestions again.